# Diversity and Molecular Evolution of Antimicrobial Peptides in Caecilian Amphibians

**DOI:** 10.3390/toxins16030150

**Published:** 2024-03-14

**Authors:** Mario Benítez-Prián, Héctor Lorente-Martínez, Ainhoa Agorreta, David J. Gower, Mark Wilkinson, Kim Roelants, Diego San Mauro

**Affiliations:** 1Department of Biodiversity Ecology and Evolution, Faculty of Biological Sciences, Complutense University of Madrid, 28040 Madrid, Spain; mariob05@ucm.es (M.B.-P.); hlorente@ucm.es (H.L.-M.); 2Natural History Museum, London SW7 5BD, UK; d.gower@nhm.ac.uk; 3Herpetology Lab, Natural History Museum, London SW7 5BD, UK; m.wilkinson@nhm.ac.uk; 4bDIV, Biology Department, Vrije Universiteit Brussel, Pleinlaan 2, 1050 Elsene, Belgium; kim.roelants@vub.be

**Keywords:** AMP, Gymnophiona, genome, transcriptome, antimicrobial activity prediction, directional selection, peptide structure modelling

## Abstract

Antimicrobial peptides (AMPs) are key molecules in the innate immune defence of vertebrates with rapid action, broad antimicrobial spectrum, and ability to evade pathogen resistance mechanisms. To date, amphibians are the major group of vertebrates from which most AMPs have been characterised, but most studies have focused on the bioactive skin secretions of anurans (frogs and toads). In this study, we have analysed the complete genomes and/or transcriptomes of eight species of caecilian amphibians (order Gymnophiona) and characterised the diversity, molecular evolution, and antimicrobial potential of the AMP repertoire of this order of amphibians. We have identified 477 candidate AMPs within the studied caecilian genome and transcriptome datasets. These candidates are grouped into 29 AMP families, with four corresponding to peptides primarily exhibiting antimicrobial activity and 25 potentially serving as AMPs in a secondary function, either in their entirety or after cleavage. In silico prediction methods were used to identify 62 of those AMPs as peptides with promising antimicrobial activity potential. Signatures of directional selection were detected for five candidate AMPs, which may indicate adaptation to the different selective pressures imposed by evolutionary arms races with specific pathogens. These findings provide encouraging support for the expectation that caecilians, being one of the least-studied groups of vertebrates, and with ~300 million years of separate evolution, are an underexplored resource of great pharmaceutical potential that could help to contest antibiotic resistance and contribute to biomedical advance.

## 1. Introduction

Antimicrobial peptides (AMPs) are a class of peptides that constitute one of the first lines of defence against pathogens, forming part of the innate immune response of many organisms [1]. They are fast-acting; have a broad spectrum of action against bacteria, viruses, fungi and parasites; and, importantly, their mode of action entails a small risk of resistance evolution in pathogens [2]. This property makes AMPs particularly interesting in light of antibiotic resistance by multiple bacterial strains, one of the most serious global health problems that humans currently face [3,4]. In addition to providing promising templates for the design of new antibiotics, AMPs have proven useful in other medical fields, such as cancer control [5], treatment of sexually transmitted diseases [6], and wound healing [7]. The general antimicrobial mechanism of AMPs involves their interaction with cell membranes, either increasing their permeability or triggering lysis and release of intracellular contents [8]. There is also increasing evidence that at least some AMPs can act (intracellularly) on nucleic acids, protein synthesis, or activities of certain enzymes [8].

AMPs are small chains of amino acids, generally between 10 and 60 residues, encoded by a variety of gene families. Most of these are well documented as encoding proteins and peptides with antimicrobial activity as their primary or sole function (e.g., cathelicidins, defensins, lysozymes; [9,10,11]). However, several genes encoding proteins or peptides with long-established non-antimicrobial functions were recently identified to serve an antimicrobial function in at least some vertebrates. In these cases, antimicrobial activity was found to be either a secondary function of the entire encoded protein or peptide (e.g., the hormones amylin and PACAP; [12,13,14]) or of one or several fragments after enzymatic cleavage (e.g., peptide fragments of histone-H2A, haemoglobin beta, and GADPH; [15,16,17]). These discoveries highlight the possibility that the true AMP diversity in many vertebrates may be underestimated and that several protein or peptide families may actually have a more widespread AMP function than currently perceived. Due to their wide diversity and multiple origins, there is no single generally accepted classification of AMPs. Commonly used classifications are based on aspects such as secondary structure (alpha-helix, beta-sheet, etc.), physicochemical properties (cationic, amphipathic, etc.), protein family assignment (histones, GADPH, etc.), activity target (against bacteria, fungi, viruses, etc.), or mechanism of action (models of membrane-pore formation, intracellular action, etc.). As a result, recent literature reviews on AMPs have often proposed alternative classifications [18,19]. Despite their disparate origins, common evolutionary mechanisms might be expected to explain their diversity due to their shared function in the fight against pathogens. Mechanisms that have been proposed to explain AMP molecular evolution include positive selection, hypermutation, domain swapping, or gene conversion [20,21,22,23,24,25,26].

Amphibians have long been recognised for the wide variety of compounds that they produce in their skin [27,28,29]. Many amphibian species generally maintain a relatively thin, moist integument that is well suited to cutaneous respiration [30], but which might expose them to a greater potential risk of infections. Hence, there is a general perception that amphibians have a great diversity of bioactive defensive compounds that are produced in the granular glands of their skin [31]. However, this perception is based almost exclusively on the study of the order Anura (frogs and toads) because the majority of known bioactive (including antimicrobial) molecules have been isolated from skin secretions of members of this order. Although some members of the order Caudata (newts and salamanders) have been investigated for the presence of alkaloids, tetrodotoxin [32,33,34], and peptides (salamandrin, cholecystokinin; [35,36]), the third extant amphibian order, Gymnophiona (caecilians) has been almost entirely ignored [37]. At least for anurans, and possibly all amphibians, it is likely that their large diversity of bioactive defensive compounds is related to their antipredator role, as suggested by König et al. [38] and later confirmed by Raaymakers et al. [39] for the case of AMPs.

Most adult caecilians are adapted to life in soil, and thus are morphologically very distinct: limbless, superficially worm-like, with small and sometimes externally invisible eyes, and a pair of sensory tentacles [40]. One family (Typhlonectidae) includes secondarily aquatic species. Caecilians are possibly the least-studied order of vertebrates [41] and are definitely the least-studied amphibian order regarding the production of bioactive compounds [37]. To date, only a few studies have addressed the skin secretions of caecilians [37,42] but none have specifically addressed AMP diversity. Nevertheless, in a recent study, Torres-Sánchez et al. [43] detected the expression of some AMPs in caecilian skin as part of a more broad-scale transcriptomic data analysis with a more general focus on skin components. Soils are often moist and thermally stable environments that harbour a high diversity of potentially pathogenic bacteria, protozoa, and fungi [44,45]. The soil-dwelling lifestyle of most adult caecilians might expose them to strong selective pressures that could have differentially modified the evolution of the pathogen defence mechanisms in this group.

In this study, we investigate the diversity and molecular evolution of the AMPs present in the genomes and transcriptomes of eight species of caecilians representing six of the ten recognised families and spread across the phylogeny of the group [46] (Figure 1). It is therefore the first study to focus exclusively on the cataloguing and detailed characterisation of the variability of the AMP repertoire in caecilians (and with greater depth and curation effort than in [43]). In addition, it is one of the first studies to identify candidate caecilian AMPs using whole-genome information, as conducted for other organismal groups by, for example, Pradhan and Engsontia [47] and Yi et al. [48].

Our main hypothesis is that, given the ~300 million years of independent evolution of caecilians [49,50] and their ecological distinctiveness, mainly due to their fossorial lifestyle, we expect to find in them a high diversity of candidate AMP molecules differentiated from those of other amphibians. Our main objectives are as follows: (1) to catalogue and characterise the diversity of potential AMPs in caecilians, (2) to analyse their molecular evolution in a phylogenetic framework, and (3) to explore the potential of in silico predictions of antimicrobial activity and three-dimensional (3D) structure modelling of the peptides found.

## 2. Results

### 2.1. Sequence-Similarity Searches

After filtering and curating BLAST sequence-similarity searches, we obtained a final number of 477 candidate AMPs in caecilian amphibians, grouped into 29 protein families or subfamilies (Table 1 and Appendix A). Most of these were identified by a significant match with sequence records in the APD3 database, except for cathelicidins and members of the cystatin-related superfamily (cystatin and kininogen) that matched entries in the UniProt database. No defensins were found in any of the studied caecilian species, neither in genomic nor in transcriptomic data. AMPs exclusive to anurans, such as magainins, CPF, XPF and bombinins, or maximins (used as control in the BLAST searches, see Section 5.1), were not found in caecilians (Appendix A).

Our screening identified transcripts and genes representing four different peptide or protein families with a known primary function as AMP. Additionally, we have identified 25 protein or peptide families that exhibit various non-AMP functions but may secondarily serve an antimicrobial role in at least some vertebrates. These include histones, chemokines, GADPH, PACAP, and cofilin, among others, with five of them appearing in all the studied species (Table 1).

### 2.2. Cataloguing of Potential AMP Sequences

Of the total number of candidate AMP sequences that were recovered in the sequence-similarity searches in each of the analysed species, fewer were found in the caecilian species for which only transcriptome data were available, especially in the case of *Geotrypetes seraphini*, *Typhlonectes natans*, and *Ichthophys kohtaoensis*. On the other hand, more candidate AMPs were found in the transcriptome of *Typhlonectes compressicauda*. Overall, *Microcaecilia unicolor* and *Rhinatrema bivittatum*, two of the three species for which both genome and transcriptome data are available, have the highest numbers of candidate AMPs (Figure 2).

As mentioned above, the candidate AMPs of caecilians obtained in the similarity searches were grouped into 29 different protein (sub)families (Table 1). We distinguished three different ways through which AMPs of the APD3 database matched proteins inferred from caecilian transcripts or genes: (i) the AMP matches along the full length of the sequence, (ii) the AMP matches a limited region of the entire sequence, not its complete length; and (iii) in the case of histone H2A, several AMPs match two different regions of the protein (see schematic representation in Appendix A). One of these regions, close to the N-terminus, matches multiple previously discovered AMPs such as hipposin, acipensins 1–5, and buforin. The other region, located 10 residues towards the C-terminus, matches acipensin 6 (see [51]).

As shown in Figure 3, the number of candidate AMPs is quite variable among the different protein (sub)families. For example, in *G. seraphini*, the higher number of some histone-derived AMPs (H2B, H2A-hipposin, H2A-acipensin 6) is noteworthy, which are followed by cofilin and lysozyme C. *M. unicolor* follows a similar pattern regarding the high numbers of histone H2B and H2A-hipposin (although followed by lysozyme C). In contrast, *R. bivittatum* has a remarkably high number of chemokines. The number of cathelicidins and cystatin-related AMPs (derived from UniProt data) are not directly comparable with the rest of the candidate AMPs (derived from APD3 data) because the specific region that might present antimicrobial activity is not annotated in the UniProt records employed for these three protein families, and therefore the entire protein sequence length was employed. Note that the information shown in Figure 3 is based exclusively on genome-derived data (which is available for only three caecilian species: *R. bivittatum*, *M. unicolor*, and *G. seraphini*) in order to provide a more standardized comparison of AMP numbers among species.

### 2.3. Phylogenetic and Directional Selection Analyses

In general, for each AMP family catalogued in this study, the phylogenetic trees recovered showed relatively low support values for many internal branches, as expected given the short lengths of the alignments. Therefore, we discuss the results in detail only for a subset of the phylogenetic trees of candidate AMPs that have some sufficiently well-supported branches to be interpreted with confidence (see Section 2.3.1, Section 2.3.2, Section 2.3.3 below). These selected trees also serve to illustrate some of the general patterns observed in most other trees, such as caecilian-specific AMP clades, and the relationships among related peptides of different protein (sub)families. The entire set of phylogenetic trees for each protein (sub)family found is available in the Appendix A. Signatures of directional selection were detected for only five candidate AMP subfamilies (Table 2): lysozyme C (three sites in two different branches of the tree), cathelicidin (two sites), PACAP (one site), and histones H2A-hipposin (one site) and H2B (one site).

#### 2.3.1. Calcitonin and Amylin Tree

Calcitonin and related amylin are widely known as an evolutionary conserved hormone in vertebrates. In various taxa however, including fishes and human, an antimicrobial activity has been reported [14], hinting at a widespread immune function activity in addition to hormone functions. The AMP amylin-BP discovered in the mudskipper *Boleophthalmus boddarti* [48] allowed for the identification of two different (but related) hormone homologs with a potential antimicrobial activity in the genomes and transcriptomes of the eight caecilian species sampled in this study: the amylin or islet amyloid polypeptide (IAPP) and the calcitonin gene-related peptide (CGRP). These two peptides were resolved as two distinct clades in the tree with high support (Figure 4). The *B. boddarti* reference sequence from the APD3 database (record AP02917) clustered with the amyloid peptide sequences, and clearly the trees imply an ancient duplication event. In the case of calcitonin, similarity searches returned only one caecilian sequence match for *T. compressicauda* (with only transcriptomic source data available), which clusters with the calcitonin sequences. In contrast, similarity searches for amylin returned matches in the genomes of *G. seraphini* and *M. unicolor* that cluster within the amylin clade.

#### 2.3.2. Lysozyme G Tree

Lysozyme G is among the protein families with well-known primary antimicrobial properties in several biological processes [52]. The highest scoring BLAST match was with the lysozyme G from the seahorse *Hippocampus abdominalis* from the APD3 database (record AP02743). The phylogenetic relationships within this AMP family were only partially resolved (Figure 5), with species from different taxonomic groups and paralogues from the same species interspersed, but without strong support. Two major clades of caecilian lysozyme G, clades 1 and 2 in Figure 5, were recovered with high support. Within these two caecilian clades, different peptide isoforms for the same species generally clustered together, and phylogenetic relationships among the caecilian genera and species were recovered with high support, mostly in agreement with previous studies of caecilian phylogeny [46]. The two caecilian lysozyme G clades are within a large clade grouping *Bombina bombina* lysozymes and some isoforms of *Xenopus laevis* and *Protopterus annectens*, excluding the lysozyme G of the seahorse, which was used as a reference sequence.

#### 2.3.3. Cofilin Tree

Cofilin or actin-depolymerizing factor (ADF) is a family of actin-binding proteins linked to the rapid disassembly of actin microfilaments [53] that was recently shown to have antimicrobial activity [54]. Phylogenetic relationships of sequences that showed similarity to the cofilin 1 of zebrafish *Danio rerio* in the APD3 database (record AP03323) are presented in Figure 6, in which three main caecilian clades (1, 2, and 3), each corresponding to a different cofilin type or paralogue, were recovered with high support and each with internal relationships largely consistent with current understanding of caecilian interrelationships [46]. However, phylogenetic relationships among these three main clades could not be resolved with confidence. In general, within clades 1, 2, and 3, different isoforms of the same species appeared clustered together.

### 2.4. Prediction of Antimicrobial Activity of Candidate Caecilian AMPs

Figure 7 summarises the results of the six activity prediction analyses conducted on the candidate AMP sequences from both genomes and transcriptomes of caecilians, grouped by protein (sub)family (detailed results for each individual candidate AMP tested are shown in Appendix A). Of the 357 caecilian peptides tested, 282 were identified as potential AMPs by at least one of the six prediction methods employed, 70 were identified as potential AMPs by four out of the six prediction methods, and 13 by all six methods. In contrast, there were 74 peptides not predicted as AMPs by any of the methods (mostly related to thymosin, vasostatin, proenkephalin, POMC, and enolase) (Appendix A).

The peptide (sub)families showing the greatest antimicrobial potential are apelin (although it consists of a single sequence of *M. unicolor*, see Figure 3), IBP, some histone H2A-derived peptides (e.g., hipposin and acipensin 6), GADPH, and neuropeptide W (Figure 7). With the exception of GADPH, these (sub)families were predicted to be AMPs lacking cytotoxicity towards human erythrocytes by at least four of the predictive methods. Cytotoxicity against mammalian cells could not be assessed for peptides longer than 30 amino acids because the existing methods in DBAASP do not allow for this. Effectiveness against Gram-positive or Gram-negative bacterial strains was predicted only for those peptides classified as AMPs by one of the two predictive methods used in AMP Scanner (see Section 5.5).

We assessed the significance of our predictions using a dataset of 1000 randomly generated peptides ranging between 5 and 150 amino acids. The methods from which we obtained the lowest number of positive hits among the randomly generated data (Appendix A) were, respectively, the RF method of Antimicrobial Peptide Scanner vr. 1, the method implemented in DBAASP, and the SVM and ANN methods of CAMPR4. Despite this, there was almost no variation among methods in the peptides scoring the highest antimicrobial potential determined using either the four most-reliable methods or all six methods. Therefore, the results from all six activity prediction methods employed were retained and are shown in Appendix A and Figure 7.

### 2.5. Three-Dimensional Structure of Caecilian GADPH-Related and Histone H2A-Related Peptides

We conducted three-dimensional (3D) reconstruction modelling of a selection of three caecilian candidate AMPs among those showing high antimicrobial potential in the activity prediction analyses (Appendix A). For each of these, and in order to provide a comparative illustration, three sequences were modelled: one from the APD3 database record (validated reference), one of a caecilian species with highest score in the prediction analyses of antimicrobial function, and one of a caecilian species with the lowest score in such tests.

The 3D structure of GADPH-derived AMPs consists of one β-sheet at each end, an α-helix in the middle, and two loops between the β-sheets and the α-helix (Figure 8). The candidate peptide from *R. bivittatum* (b in Figure 8) scored well in the predictive analyses and also shows high similarity in structure to the AMP in the APD3 database. Although the structure of the three peptides is quite similar, the peptide from *Caecilia tentaculata* (c in Figure 8), which obtained a lower score in the predictive analyses, appears to have a higher dissimilarity in particular regions with the AMP from the database. This peptide has a two-amino acid (LV) insertion in the α-helix region. Overall, the 3D structure is generally maintained for both peptides despite a few nonconserved positions being present in them.

The structure of histone H2A-acipensin 6 is relatively simple, as expected from the small size of the molecule (21 amino acids) (Figure 9). It consists mainly of an α-helix with an uncoiled C-terminus, which is the most variable region of the protein. The α-helix is well-conserved in all three peptides, though there are some disparities in the C-terminus region (the start and end regions of proteins are usually modelled more inaccurately). Despite poorer 3D model prediction, the alignment shows higher similarity between the *R. bivittatum* peptide and the AMP in the APD3 database (Figure 9).

The 3D structure of the histone H2A-hipposin (Figure 10) contains a short α-helix at approximately one third of the sequence, followed by a longer α-helix and another short α-helix close to the C-terminus. The peptide with predicted good antimicrobial activity (*G. seraphini*, b in Figure 10) lacks the C-terminal α-helix. Conversely, the peptide with a low score in the antimicrobial prediction (that of *T. compressicauda*, c in Figure 10) is remarkably similar in structure to the H2A-hipposin in the APD3 database.

## 3. Discussion

### 3.1. Diversity of Candidate AMPs in Caecilians

Our analysis of the genomic and transcriptomic data of eight species of caecilians—from across the phylogeny of Gymnophiona (Figure 1)—yielded a high diversity of candidate AMPs in 29 different protein (sub)families (Table 1, Figure 2 and Figure 3). In general, our results showed a notable degree of sequence conservation among caecilian species. We found at least five different AMP (sub)families that are present in all studied species, as well as anurans and lungfish (Table 1 and Appendix A). These well-conserved protein/peptide (sub)families correspond to genes for which antimicrobial activity is not known as a primary function. Even so, the candidate AMPs show sequence variation among species. Antimicrobial assays will be needed to investigate whether this variation implies an alteration in specificity or function. Nevertheless, it is likely that these molecules can evolve relatively rapidly—for example when an amphibian engages in an evolutionary arms race with a pathogen—and sometimes a change in one amino acid can drastically modify the activity of an AMP [23].

In our similarity searches, we have found (i) sequences similar to known AMPs that have a primary or unique function as AMPs (e.g., lysozymes, LEAPs, etc.; [9,55,56]); and (ii) sequences of proteins whose originally established function is not antimicrobial, but for which antimicrobial activity has been reported in some studies [16,17,57,58]. Of these latter, some show antimicrobial activity only for cleaved fragments (such as histone-H2A or GADPH). The primary function of some of these proteins is well known, such as DNA packaging in the case of histones or the role of the enzyme GADPH in glycolysis. These proteins are widely distributed and some of them are constitutively expressed in virtually all tissues (and cells). Therefore, finding these sequences does not guarantee that they may serve an additional antimicrobial function. Proteomic or peptidomic analyses will be required to prove that caecilians (or other taxa) indeed process these peptides as proper AMPs. Nonetheless, it is remarkable that some of these peptides (such as those from histone-H2A) have scored among the highest as candidate AMPs in the predictive analyses (see Section 3.3). In general, proteins like histones have a high degree of sequence conservation (in principle related to their primary function), and this may have facilitated the maintenance of an antimicrobial capacity even in species where these proteins are not involved in immune response anymore.

Four protein families with antimicrobial activity as a primary function were recovered in the studied caecilians, namely cathelicidins, LEAP, and lyzozymes C and G (Table 1). It is remarkable that we could not find any match to defensins (not even distant matches) neither in genome nor in transcriptome data. Therefore, it appears that, during the course of evolution, caecilians have lost defensins. On the other hand, several of the peptides found with possible secondary antimicrobial functions correspond to hormones or neuropeptide families (adrenomedullin, DBI, vasostatin, neuropeptides W and YY) that indeed may be interesting from a toxicological perspective, acting as anti-predator toxins instead of AMPs by targeting the hormone receptors of a predator.

Our results also show that more candidate AMP-encoding sequences were found in the caecilian genomes than in the transcriptomes (Figure 2). Transcriptomes represent only genes expressed at a particular time and location (cell or tissue), so unexpressed genes would not be included. In this sense, the low number of sequences obtained from the transcriptomes of *G. seraphini*, *T. natans*, and *I. kohtaoensis* is noteworthy (Figure 2). For these species, many of the matches were obtained with sequences for which we could not straightforwardly find a clear and complete ORF. Such artefactual sequences had to be manually realigned to identify the correct reading frame, though this procedure proved ineffective for some sequences. Moreover, we might have encountered spurious results due to the suboptimal performance of the similarity search algorithms or the possibly low quality of some assemblies. We therefore took a conservative approach and discarded all potentially spurious sequences of *G. seraphini*, *T. natans*, and *I. kohtaoensis* from the main analyses.

In general, a large number of repeated sequences or duplicates were found in all studied species (caecilians and reference anurans and lungfish) during the curation and filtering steps (see Section 5.3). In fact, one of the datasets with the highest proportion of duplicates was that of the lungfish, whose genome is known to be large and very repetitive [59]. In contrast, caecilians have considerably smaller genomes [60]; hence, explaining the relatively high proportion of duplicates in them is not straightforward. AMPs are small peptides whose effectiveness is often concentration-dependent [8,61], so it is therefore plausible that a high number of duplications of these genes would allow the organism to synthesise many copies in a short period of time. The presence of duplicate genes could also enable neofunctionalization processes [62,63], where the original function of the peptide may be conserved in one of the copies while the other(s) may undergo increased mutation and evolve a new function.

The different caecilian sequences that were recovered as candidate AMPs could be classified into 29 different protein (sub)families (Table 1). There is no unified classification for these molecules, and different authors use different classifications [8,19]. In our case, we have classified them into protein or already known AMP families or subfamilies, as performed in similar previous studies [48]. Overall, the pattern in terms of number of sequences recovered in most caecilian species was somewhat similar across these 29 families (Figure 3), with the AMP families that comprised most sequences in *G. seraphini* also being the most numerous in *R. bivittatum* and *M. unicolor*. This pattern may be explained by the preservation of genes that duplicated in a common ancestor of caecilians. Also noteworthy is the high number of chemokine sequences (and, in a lower proportion, of lysozyme G) in *R. bivittatum*, which may reflect adaptation in response to an increased need for immune protection. A possible explanation could be related to the fact that this caecilian species has an aquatic larval stage followed by a (sub)terrestrial adult stage [46]. Perhaps the AMP repertoire represented by these two protein families expanded adaptively, in response to a pathogen threat specific to one of its two life history stages, as has been suggested for other amphibians [64]. We also observed a high number of candidate AMP sequences in the frog *X. laevis* (used as reference). This observation may be explained by the tetraploid genome of this species [65], although Mechkarska et al. [66] have observed that polyploidy does not appear to increase the diversity of expressed AMPs in other species of the family Pipidae. Conversely, in the case of *I. kohtaoensis*, the low number of candidate AMPs reported here is likely related to the fact that skin tissue was not included in the generation of the original transcriptomic data [67]. Thus, an important component of the AMP diversity was likely not captured and is missing from the source data used here for the similarity searches.

### 3.2. Molecular Evolution of Candidate AMPs

In general, the alignments of candidate AMP sequences have limited ability to resolve the phylogenetic relationships, and inferred trees only partially reflect the known species phylogenies (see Appendix A). This is likely due to their short sequence length, high variability, and the fact that AMPs often undergo convergent evolution [23]. In addition, the patterns observed for the different candidate AMPs appear to suggest frequent events of gene duplications and losses in caecilians, as found for other amphibians and more distantly related taxa [21,47].

Among the phylogenetic results highlighted here, the calcitonin and amylin tree (Figure 4) showed two well-supported sister clades for calcitonin and amylin (or islet amyloid polypeptide). The calcitonin clade contains only one caecilian sequence, *T. compressicauda*, for which only transcriptomic source data are available. In contrast, the amylin similarity searches contain sequences of *G. seraphini* and *M. unicolor*. No sequences homologous to calcitonin were found in these latter two species, suggesting that perhaps their calcitonin sequences are too diverged to be identified by similarity searches, or that they are absent.

In the case of lysozyme G (Figure 5) and cofilin (Figure 6) trees, several paralogues are recovered. In both cases, there are well-supported clades of caecilian candidate AMPs (often with long branches) that suggest a higher differentiation in those peptides—this pattern was also observed in some of the other trees not illustrated in the main text (Appendix A). Conversely, it is likely that such long branches negatively affect resolution and support in the rest of the tree. In the case of the cofilin tree, there is a larger number of sequences for caecilians than in the other sampled taxa. This sequence enrichment, plus the presence of the three well-differentiated clades, might indicate exclusive events of duplication in caecilians that might be explained by adaptation to their mostly soil-dwelling lifestyle [68], which may expose them to a distinct battery of microorganisms compared to other amphibians.

Molecules that act in defence against pathogens such as AMPs represent some of the best-known examples of directional or balancing selection [20,69,70]. Directional selection favours the change and diversification of AMPs that are subject to selective pressures that may be imposed by arms races in the fight against pathogen infections, and balancing selection tends to maintain this diversity [20,24]. We detected directional selection signatures in five AMP candidates, namely lysozyme C, cathelicidin, PACAP, and histones H2A-hipposin and H2B (Table 2). The existence of directional selection signatures might indicate positions that can evolve rapidly and which could be related to pathogen-fighting defence mechanisms. This may particularly be the case of those AMPs that have antimicrobial function only, such as lysozyme C and cathelicidins. Instead, proteins or peptides with a secondary antimicrobial function (such as histones or PACAP) may be subject to adaptive conflicts between an antimicrobial function and its alternative function. In such cases, directional selection to optimise the antimicrobial function will be constrained by selection to retain the other function. It has been demonstrated that gene duplication may release genes of adaptive conflicts, allowing both daughter genes to diversify in function. This ‘Escape from adaptive conflict’ scenario [71] has previously been shown to yield functionally diversified genes and, in the context of AMPs, may explain the presence of multiple paralogues in some of the studied caecilians.

### 3.3. In Silico Activity Prediction of Antimicrobial Properties

The search for effective treatments to deal with pathogens has become a pressing challenge due to the constant evolution of microorganisms and the development of new resistance mechanisms [72]. In this regard, artificial intelligence (AI)-based tools are becoming particularly relevant and effective to accelerate the discovery of potential new bioactive compounds [73]. In the context of the present study, the use of activity prediction tools becomes particularly important for those AMP candidates that do not have a clear antimicrobial function but are homologous to proteins or peptides for which antimicrobial activity has been reported as a secondary function. In fact, by using different activity prediction methods, we were able to determine that some of the candidate AMPs found in caecilians have significant antimicrobial potential (Figure 7). For their possible therapeutic use, it is important to know not only whether the peptide has antimicrobial properties, but also whether it can be harmful to human cells, which is why we also tested in silico for cytotoxicity against human erythrocytes.

A total of 62 caecilian peptides were predicted to be AMPs by all or most (four out of six) of the methods used and showed low activity against human erythrocytes (e.g., most GADPH peptides were excluded of this count because they exhibited such cytotoxicity; see Appendix A), making them the most interesting candidates. These peptides are apelin, IBP, histone H2A-derived peptides (hipposin and acipensin 6), and neuropeptide W. As mentioned in Section 3.1, these peptides have a primary function other than antimicrobial activity, but in light of our results, they could be (i) playing some kind of antimicrobial role (even if secondary) in the studied caecilian species or (ii) despite not having antimicrobial function in the studied species, could have at least particular domains with antimicrobial activity due to their increased conservation because they are widely distributed and carry out vitally important functions in the organisms. In contrast, some of the candidate sequences from the AMP families with exclusive or primary antimicrobial function (lysozymes C and G, LEAP) scored lower overall in the predictive analyses (i.e., they were predicted as AMPs by fewer methods; Figure 7). This could be explained by these AMPs having lost their antimicrobial properties in the caecilian species studied (which seems very unlikely, at least in the case of lysozymes) or because they have not been correctly scored by predictive methods due to their larger size. In general, these predictive tools are trained with data mainly represented by short-length AMPs (<100 residues).

As an additional check, we tested to some extent the significance of the activity prediction methods employed using randomly generated data. There are variations in the number of positive hits among the randomly generated data by each of the methods but, in general, these discrepancies do not significantly impact the results. A possible explanation of this might be that positive hits among the randomly generated data could result from the inclusion of amino acid sequences that differ substantially from the antimicrobial peptides that these AI algorithms were trained on, something that is crucial for the performance of AI methods [73].

### 3.4. Structural Modelling of Candidate AMPs

The function and mechanism of action of AMPs is intrinsically linked to their spatial secondary structure. For example, many AMPs conform to an α-helical structure that interacts with the bacterial membrane in such a way that its size, amino acid sequence, charge, and hydrophobicity affect the antimicrobial spectrum and activity. However, the vast majority of discovered AMPs do not have a known 3D structure. Traditional physical-chemical methods for determining the structure of peptides, such as X-ray crystallography or microscopy, are accurate, but laborious and expensive [74].

Recent advances in AI have allowed for the development of tools to predict the 3D structure of peptides or proteins with high accuracy in a (cost-effective) in silico context. These tools bring the potential to characterise AMPs and make predictions of structure, broader antibacterial activity, and more generally, their biological function. The predicted structures can inform about structure–function relationships (e.g., how chemical changes can affect antimicrobial activity), and coupled with antimicrobial activity predictions can be informative for example in selecting peptides identified in silico as candidate AMPs for in vitro validation [74]. In this context, the in silico characterisation of AMP 3D structures is certainly a cost-effective initial step for shedding light in the mechanisms of action of these diverse molecules. In fact, 3D-modelling techniques are nowadays used regularly in drug design, protein and peptide engineering, and targeted mutagenesis [75].

The 3D structures of the three candidate AMPs analysed, of GADPH, histone H2A-acipensin 6, and histone H2A-hipposin (depicted in Figure 8, Figure 9 and Figure 10), generally resemble those of the corresponding AMPs from the APD3 database, for which antimicrobial activity has been proven. In the case of GADPH, the peptides recovered in caecilians with good antimicrobial activity potential show higher structural similarity to the database AMP than those with lower potential (Figure 8). In contrast, the histone H2A-hipposin peptide (Figure 10) shows the opposite trend, with the caecilian sequence having good antimicrobial potential but with 3D-conformational differences to the database AMP. This may indicate a change in the mechanism of action, which is strongly related to the 3D conformation [75,76] or a change in functionality. These peptides with distinct 3D conformations may be relevant because they might represent novel AMPs exclusive to caecilians, potentially attracting pharmacological interest.

Histone H2A-acipensin 6 (Figure 9) has a straightforward α-helix structure, suggesting a probable mechanism of action as a membrane disruptor, which is the main mechanism of action of AMPs [8,77]. The α-helix structure is the most effective conformation when interacting with the bacterial membrane [78]. These peptides are likely to function by aggregating with other acipensins and forming pores in the pathogen membrane [79]. In many AMPs, α-helix structures are commonly followed by one or several β-sheets, as in our reconstructed GADPH 3D model (Figure 10), which are usually stabilized with disulphide bonds [80].

## 4. Conclusions

Amphibians are the major vertebrate group in which most AMPs have been characterised, although practically all the information available to date is exclusively for representatives of the order Anura (frogs and toads). For the first time, we made a detailed catalogue and characterisation of AMPs and their molecular evolution in a sample of caecilians (Gymnophiona), the least well-known order of amphibians, and possibly of land vertebrates.

Through detailed analyses of the whole-genome and transcriptome data available for caecilians, we identified a large diversity of peptide sequences with antimicrobial potential (candidate AMPs), classified into 29 different protein families or subfamilies. Of these, 4 correspond to peptides primarily exhibiting antimicrobial activity and 25 potentially serve as AMPs in a secondary function. Directional selection signatures were detected in five candidate AMPs of caecilians, which may indicate processes of adaptation to different selective pressures imposed by the arms race against the specific pathogens to which they are subjected. Changes in function might correlate to conformational changes in the 3D structure, as suggested by some of our candidate AMP reconstructions.

Using in silico activity prediction tools, we found that 62 of the 357 caecilian candidate AMPs tested exhibit promising antimicrobial potential. This opens a field of pharmaceutical and biomedical interest that deserves to be explored in future studies. A reasonable next step may be conducting experimental (in vitro) antimicrobial assays to study the activity and function of these candidate AMPs empirically. This would enable confirmation or rejection of the in silico predictions made here, and could also yield more accurate insights into their actual inhibitory spectra to different bacterial strains or into their chemical and immunomodulatory properties.

## 5. Material and Methods

### 5.1. Data Retrieval

Annotated protein sequences from whole-genome sequence data were downloaded from GenBank for three species of caecilians: *R. bivittatum* published by Rhie et al. [81] and *M. unicolor* and *G. seraphini* published by Ovchinnikov et al. [82]. In addition, transcriptomic data were obtained for eight caecilian species, three of them coincident with the species with available genomes plus *Microcaecilia dermatophaga*, *C. tentaculate,* and *T. compressicauda* published by Torres-Sánchez et al. [83], *T. natans* and *T. compressicauda* published by Irisarri et al. [49], and *I. kohtaoensis* published by Lu [67]. These species represent six out of the ten currently recognised caecilian families [68,84] and a reasonable-though-incomplete sample from across the caecilian tree (Figure 1) [46]. In order to make comparisons of the results obtained with other amphibians, annotated protein sequences from genome data were also downloaded from two anurans, the African clawed frog, *X. laevis*, published by the International *Xenopus* Genome Consortium (https://www.xenbase.org/, accessed on 9 January 2023), and the European fire-bellied toad, *B. bombina*, published by Rhie et al. [81]. These genomes were also used as a control to try to maximise the correct identification of potential AMPs in the caecilians, given that anurans encompass nearly all of the diversity of AMPs discovered in amphibians thus far. Finally, annotated protein sequences from genome data were also downloaded for the African lungfish, *P. annectens*, published by Wang et al. [59], providing an phylogenetically more distantly related outgroup to caecilians and to land vertebrates in general. The anuran and lungfish outgroups were employed as references in the phylogenetic and directional selection analyses (see Section 5.4).

Two types of data retrievals (and subsequent similarity searches; see Section 5.2) from AMP datasets were conducted. First, all the sequences of AMPs listed in the Antimicrobial Peptide Database (APD3; [85]; as of 20 December 2023) were downloaded. A total of 3934 protein sequences were obtained, and a custom database of AMPs sequences was constructed with them for subsequent analyses. Second, an additional dataset encompassing amphibian defensins and all candidate cathelicidin AMPs reported by Torres-Sánchez et al. [43] (see Appendix A of [43]) was also downloaded, corresponding to entries in the UniProt database. This later database allowed us to more specifically address the initial absence of matches for these AMPs (with genuine primary antimicrobial function) in the searches based exclusively on the APD3 data (see Table 1).

### 5.2. Identification of Antimicrobial Peptides

In order to identify AMPs present in the retrieved protein and transcriptomes files, we conducted BLAST (similarity) searches [86] using the BLASTP and the TBLASTN tools, respectively. The custom AMP database was used as query against the genomes and transcriptomes of interest, using the workflow and pipeline described in [87]. This pipeline allowed us to effectively retrieve protein files derived from genomic data from the online repositories. Additionally, it facilitated the processing and mining of genomic and transcriptomic data for molecular evolution purposes, a methodology recently applied to aquaporin genes in amphibious fishes [88].

AMPs are small and highly variable molecules; thus, we used previously described anuran antimicrobial sequences from *X. laevis* (e.g., magainins/CPF/XPF) and *B. bombina* (e.g., bombinins/maximins) as a control to determine whether our searches were too stringent or not. Similarity searches were conducted using an E-value of <10^−5^, enabling the successful retrieval of these control anuran AMPs. In order to minimise the recovery of false positives, stringent inspection and curation of the BLAST results were applied, including filtering out clear alignment artefacts, very distant homologues (mostly derived from non-vertebrate APD3 records), or spurious hits (e.g., representing a distant paralogue to a protein with AMP activity but not being an AMP itself). Although the BLASTP results facilitated the direct extraction of amino acid AMPs from the protein datasets, for those obtained from TBLASTN (for transcriptome sequences), it was necessary to translate the original nucleotide sequences of the matched entries.

### 5.3. Alignment, Debugging, and Duplicates Removal

BLAST searches facilitated the classification of matched sequences from the protein and transcriptome files into one of the protein or AMP families documented in the literature (see Table 1). For histone H2A, two files were generated (see Section 2.2 in Results). Only those protein families with at least one caecilian sequence were included for further analysis (see Table 1 and Figure 3). Once clustered, these sequences were aligned using MAFFT v7.520 [89]. In each alignment, we added the sequences of the AMP records from the APD3 database that had scored best in the similarity searches. This allowed us to assess the similarity between the database AMP sequence and the matched sequences. Additionally, this also served to establish a reference for the potential specific region of the candidate AMP within the recovered sequences. These alignments were manually inspected using GENEIOUS PRIME v2023.1.2 (https://www.geneious.com/, accessed on 13 June 2023), and sequences that contained long gap regions that did not align correctly or that did not appear to match the reference AMP were removed. Finally, only the region corresponding to the candidate AMP was extracted from the resulting alignments and used in subsequent analyses.

Many of the extracted sequences were identical due to (i) duplication phenomena in the genome and the conservation of the sequences, (ii) different isoforms of the same protein annotated independently, or (iii) possible errors in the assembly of the genomic data. These repetitions did not provide additional or relevant information for some of the analyses (e.g., phylogenetic or antimicrobial activity prediction analysis). Therefore, we removed the duplicated sequences and retained only unique sequences for each identified type of AMP per species.

### 5.4. Phylogenetic and Directional Selection Analyses

The resulting amino acid sequence alignments were analysed using maximum likelihood (ML; [90]) phylogenetic inference. The analyses were conducted with IQ-TREE [91,92] using an automatic selection of best-fit substitution model with ModelFinder [93]. Branch support was evaluated using 1000 replicates of SH-aLRT and Ultra-Fast Bootstrap (UFBoot) [94].

In order to assess the presence of signatures of directional selection, we used the FADE method (https://www.hyphy.org/methods/selection-methods/#fade, accessed on 7 August 2023) implemented in HyPhy v2.5.33 [95], which is based on a Bayesian approach and performed on a protein alignment. For each site in the alignment, FADE tests whether a specified set of foreground branches shows a substitution bias towards a particular amino acid, compared to background branches. We applied the test to all candidate AMP families found in caecilians, and both their respective trees and alignments were refined as follows. For clades that exclusively clustered more than one sequence of the same species (with branches not significantly dissimilar), only one was retained. Additionally, sequences with less than 90% of the entire sequence were eliminated. After removal of these partial sequences, positions containing gaps were also discarded for the same reason. In all cases, the caecilian sequences were selected as foreground branches. Statistical significance in FADE is assessed with Bayes Factors. Values of Bayes Factors above or equal to 100 indicate strong evidence that a particular site is evolving under directional selection.

### 5.5. In Silico Prediction of Antimicrobial Properties

In order to predict in silico the antimicrobial activity of the candidate AMPs of caecilians, different methods based on artificial intelligence (AI) were employed, such as RF (Random Forest), SVM (Support Vector Machine), and ANN (Artificial Neural Network), implemented in the CAMPr4 web server (http://camp.bicnirrh.res.in, accessed on 16 August 2023; [96]); RF and Earth, were both implemented in Antimicrobial Peptide Scanner (AMP Scanner) vr. 1 (https://www.dveltri.com/ascan/v1, accessed on 16 August 2023; [97]). With the latter tool, we also calculated predictive values for the effectiveness of each peptide against Gram-positive and Gram-negative bacterial strains. Additionally, we also used the method implemented in DBAASP (https://dbaasp.org, accessed on 18 August 2023; [98]) specifically designed to predict linear and cationic AMPs. The haemolytic activity of each peptide against human erythrocytes was also calculated in order to assess possible cytotoxicity on mammalian cells [99]. Predictive analyses of antimicrobial activity could not be conducted on the cathelicidin and cystatin-related sequences retrieved from UniProt because their lengths were greater than 150 residues in all cases. Hence, the final number of caecilian peptides that could be tested for antimicrobial activity was 357.

Prior to these predictive analyses, we evaluated in silico the significance of the six methods employed. A dataset with 1000 randomly generated peptides with lengths of 5–100 amino acids was simulated using a custom script. In principle, it is expected that these simulated peptides do not show any antimicrobial properties because of their random nature (or it is at least unlikely). We used this random peptide dataset to evaluate the percentage of positive hits among the randomly generated data that each of the prediction tools can produce and applied it as a measure of their significance and reliability (Appendix A).

### 5.6. In Silico Prediction of the 3D Structure of Candidate AMPs

We used AlphaFold [100] as implemented in the ColabFold v1.5.3 web server [101] to model the 3D structures of several caecilian candidate AMPs (a selection of those showing high antimicrobial potential in the activity prediction analyses, Appendix A). For each AMP family, three sequences were modelled: one from the APD3 database, one with the highest score in the prediction analyses of antimicrobial function, and one with the lowest score in such tests. We used 12 recycles per analysis, as per a preliminary assessment. It is convenient not to underestimate this parameter, given that Colabfold automatically stops the analysis when a specific level of confidence in the structure is reached [101].

After obtaining the 3D structures (PDB files) for the two caecilian candidate AMP sequences in each family, they were superimposed onto the 3D structures of the AMPs from the database using ChimeraX v1.6 [102,103]. Additionally, to assess the degree of similarity among residues in different regions, we employed root-mean-square deviation (RMSD) with the “Render by attribute” option. This is the distance between the atoms of two overlapping molecules. In addition, we constructed alignments for the studied peptides, showing their secondary structure using the ESPript v3.0 web server [104].

## Figures and Tables

**Figure 1 toxins-16-00150-f001:**
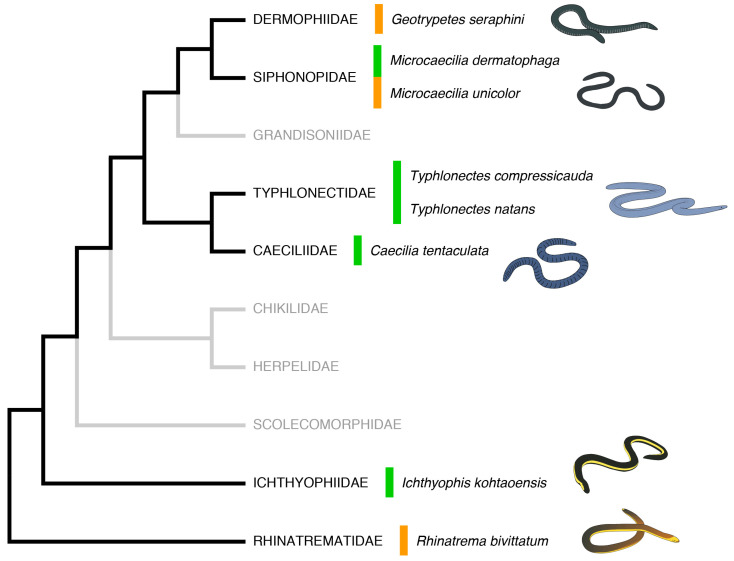
Phylogenetic tree of caecilian families highlighting those with representative genomic and/or transcriptomic data used in this study (black branches). Families without available data are shown in grey. The studied species are shown and the colour of each bar indicates the availability of genomic and transcriptomic (orange) or only transcriptomic (green) data.

**Figure 2 toxins-16-00150-f002:**
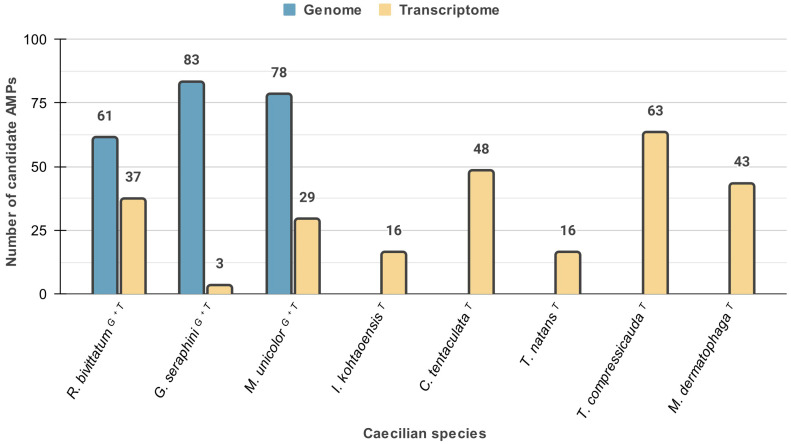
Count of candidate AMPs in the studied caecilian species, segregating information based on genomic (blue bars) and transcriptomic (gold bars) data. Only three caecilian species (indicated with G + T superscripts) have genomic and transcriptomic data available. The other five caecilian species only have transcriptomic data available (indicated with T superscripts).

**Figure 3 toxins-16-00150-f003:**
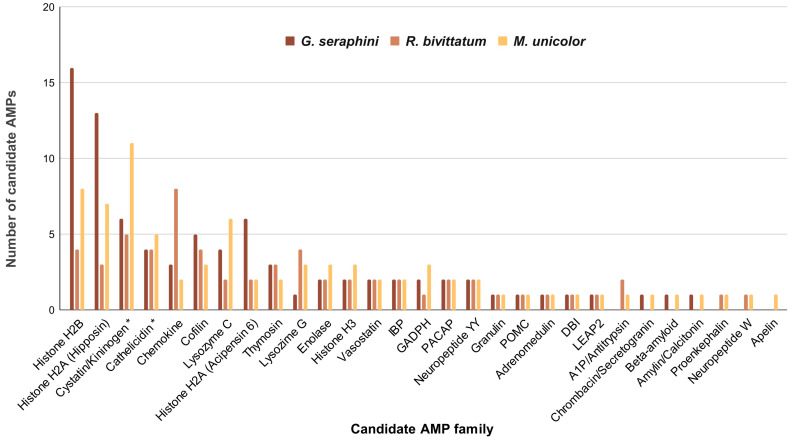
Number of pooled AMPs from genomes displayed by caecilian species and protein family. The information shown is based on genome-derived data exclusively (which is available for only three caecilian species) in order to provide a more standardized comparison of AMP numbers among species. The different colours indicate the different species under study, as shown in the legend. Asterisks denote BLAST matches against UniProt records.

**Figure 4 toxins-16-00150-f004:**
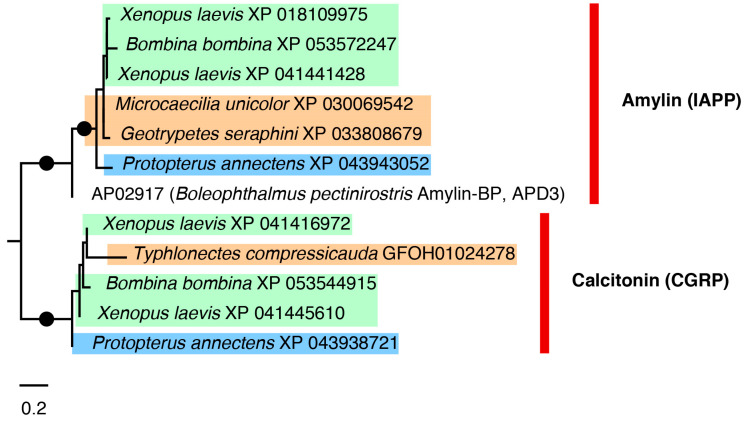
ML phylogeny of calcitonin gene-related peptide (CGRP) and amyloid or islet amyloid polypeptide (IAPP) sequences using best-fit Dayhoff + G4 substitution model. Black dots on branches represent support values above 80% and 95% for SH-aLRT and UFBoot, respectively. Colour highlights of species names are as follows: orange for caecilians, green for anurans, and blue for the lungfish.

**Figure 5 toxins-16-00150-f005:**
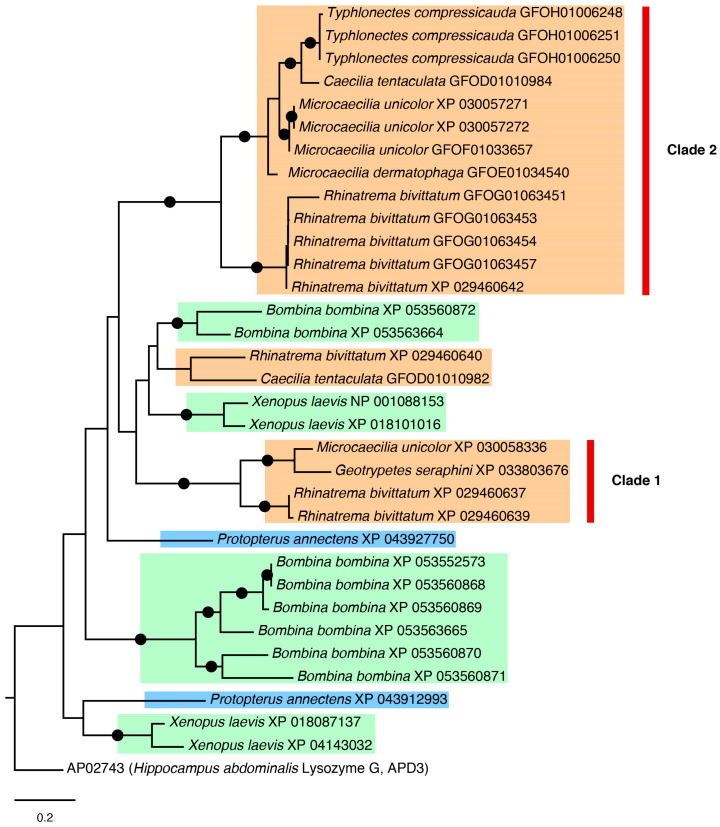
ML phylogeny of lysozyme G sequences using best-fit WAG + G4 substitution model. Black dots on branches represent support values above 80% and 95% for SH-aLRT and UFBoot, respectively. Clades 1 and 2 discussed in the text are indicated with red bars. Colour highlights of species names are as follows: orange for caecilians, green for anurans, and blue for the lungfish.

**Figure 6 toxins-16-00150-f006:**
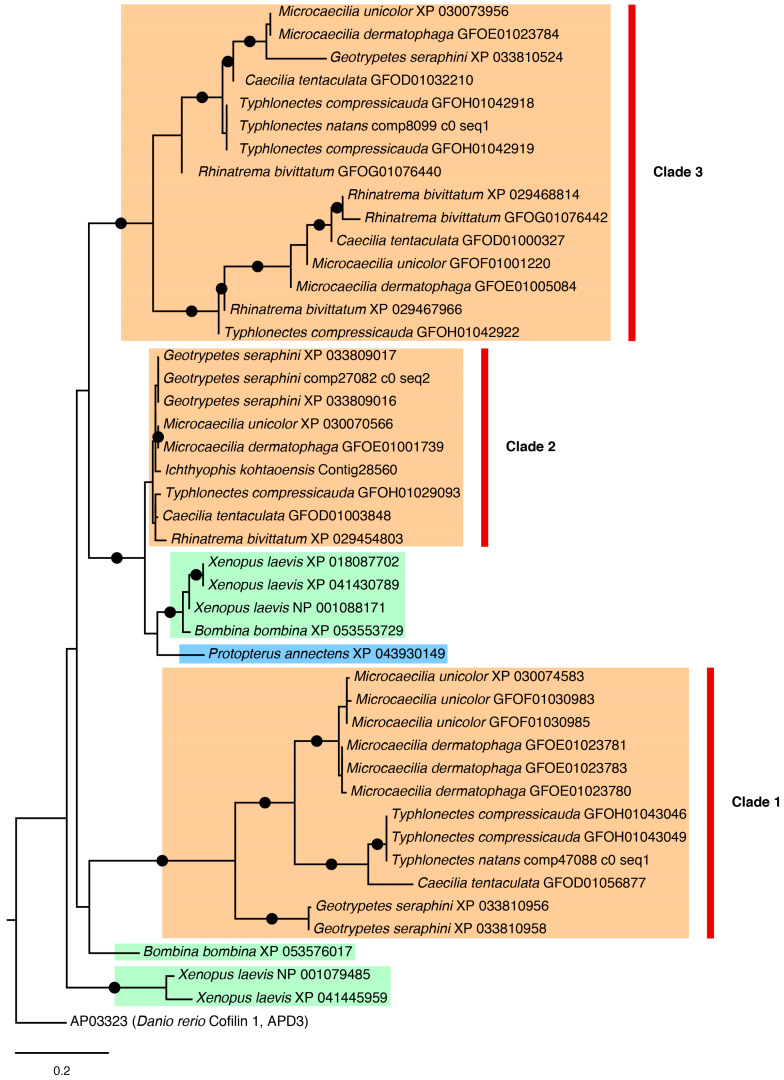
ML phylogeny of cofilin sequences using best-fit LG + G4 substitution model. Black dots on branches represent support values above 80% and 95% for SH-aLRT and UFBoot, respectively. Clades 1, 2, and 3 discussed in the text are indicated with red bars. Colour highlights of species names are as follows: orange for caecilians, green for anurans, and blue for the lungfish.

**Figure 7 toxins-16-00150-f007:**
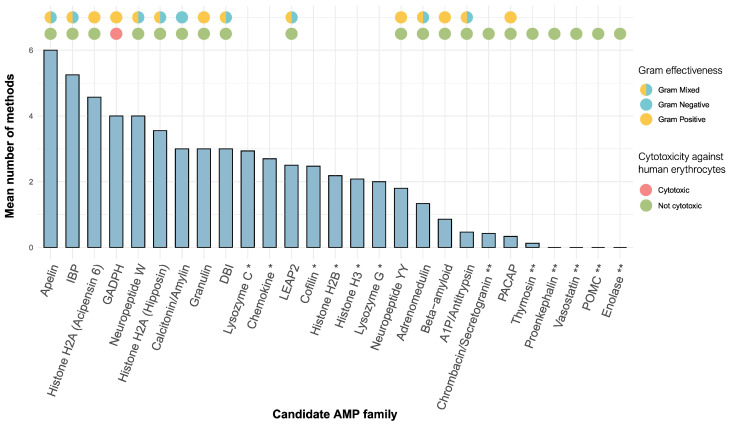
Summary of the predicted antimicrobial characteristics of the candidate caecilian AMPs catalogued in this study. The Y-axis shows the mean number of different activity prediction methods that indicate antimicrobial capacity in the peptides analysed. Effectiveness against Gram-positive and/or Gram-negative bacteria and cytotoxic activity against human erythrocytes is shown as colour circles above each bar. Candidate AMPs marked with an asterisk (*) are peptides for which neither Gram effectiveness nor cytotoxicity could be calculated because they are >100 amino acids in length and they were not classified as AMPs by the AMP Scanner method. Those candidate AMPs marked with two asterisks (**) are peptides for which only Gram effectiveness could not be calculated (not classified as AMPs by the AMP Scanner method).

**Figure 8 toxins-16-00150-f008:**
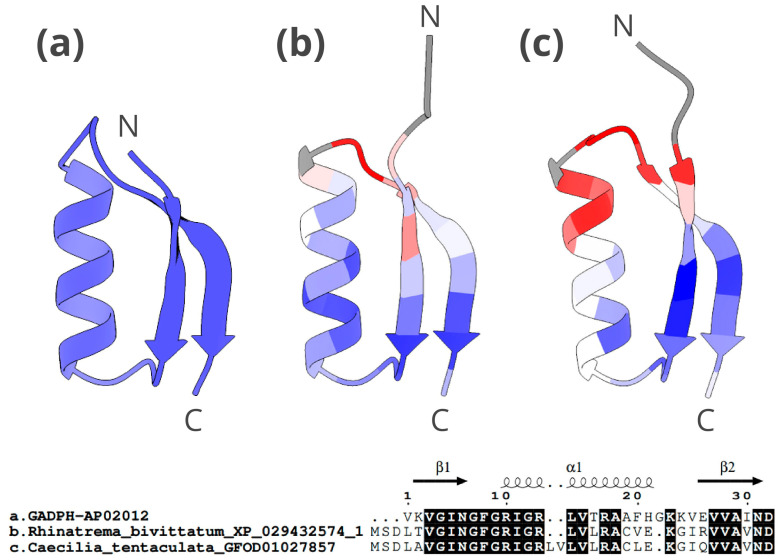
The 3D structure and alignment of GADPH-derived AMPs for *R. bivittatum* and *C. tentaculata*. The upper part of the figure shows: (**a**) peptide from APD3 (antimicrobial activity validated), (**b**) caecilian candidate AMP with high antimicrobial potential based on in silico predictive analyses (*R. bivittatum*), (**c**) caecilian candidate AMP with low antimicrobial potential based on in silico predictive analyses (*C. tentaculata*). The latter two follow a colour scale ranging from blue (highest similarity) to red (lowest similarity) related to the c-α atom deviation (RMSD) with respect to APD3 AMP. The N-terminus and C-terminus ends are marked with an N and a C, respectively. The lower part of the figure shows the alignment of the three peptides analysed. The conserved regions in the three peptides are highlighted in black. The structural motifs are shown above the alignment.

**Figure 9 toxins-16-00150-f009:**
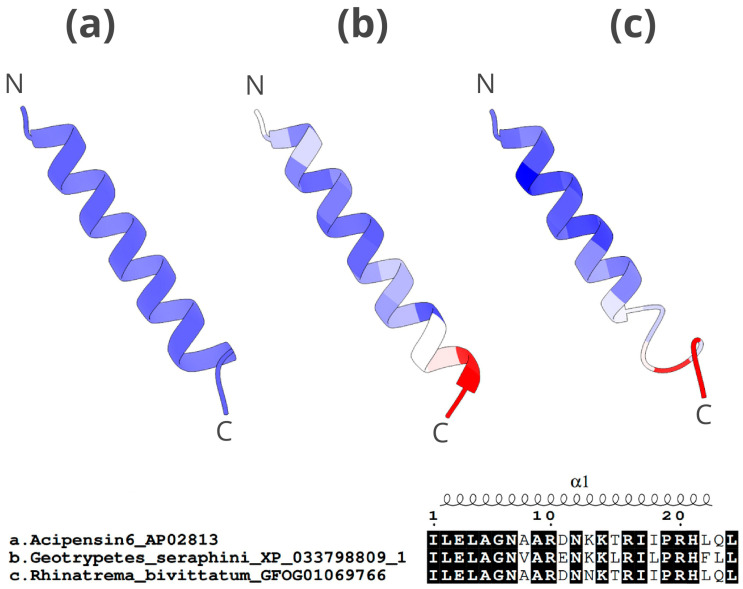
The 3D structure and alignment of histone H2A-acipensin 6 candidate AMPs for *G. seraphini* and *R. bivittatum*. The upper part of the figure shows: (**a**) peptide from APD3 (antimicrobial activity validated), (**b**) caecilian candidate AMP with high antimicrobial potential based on in silico predictive analyses (*G. seraphini*), (**c**) caecilian candidate AMP with low antimicrobial potential based on in silico predictive analyses (*R. bivittatum*). The latter two follow a colour scale ranging from blue (highest similarity) to red (lowest similarity) related to the c-α atom deviation (RMSD) with respect to APD3 AMP. The N-terminus and C-terminus ends are marked with an N and a C, respectively. The lower part of the figure shows the alignment of the three peptides analysed. The conserved regions in the three peptides are highlighted in black. The structural motifs are shown above the alignment.

**Figure 10 toxins-16-00150-f010:**
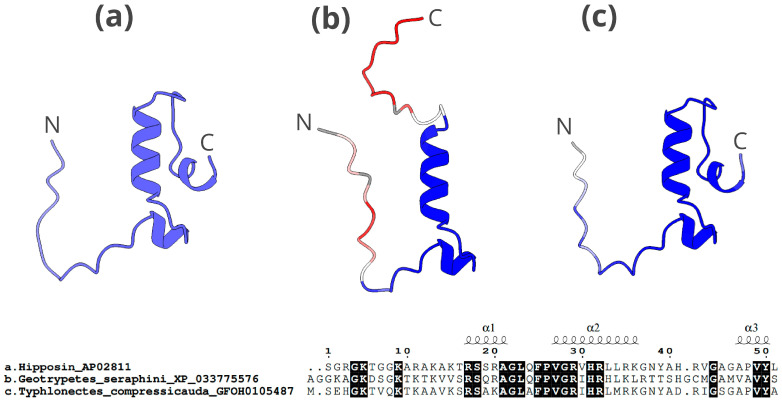
The 3D structure and alignment of histone H2A-hipposin candidate AMPs for *G. seraphini* and *T. compressicauda*. The upper part of the figure shows: (**a**) peptide from APD3 (antimicrobial activity validated), (**b**) caecilian candidate AMP with high antimicrobial potential based on in silico predictive analyses (*G. seraphini*), (**c**) caecilian candidate AMP with low antimicrobial potential based on in silico predictive analyses (*T. compressicauda*). The latter two follow a colour scale ranging from blue (highest similarity) to red (lowest similarity) related to the c-α atom deviation (RMSD) with respect to APD3 AMP. The N-terminus and C-terminus ends are marked with an N and a C, respectively. The lower part of the figure shows the alignment of the three peptides analysed. The conserved regions in the three peptides are highlighted in black. The structural motifs are shown above the alignment.

**Table 1 toxins-16-00150-t001:** Summary of candidate AMPs found in caecilians grouped by AMP or protein (sub)family.

	*R. bivittatum* ^G+T^	*I. kohtaoensis* ^T^	*C. tentaculata* ^T^	*T. natans* ^T^	*T. compressicauda* ^T^	*M. unicolor* ^G+T^	*M. dermatophaga* ^T^	*G. seraphini* ^G+T^
**AMP families (primary antimicrobial function)**								
LEAP2	X	X	X			X		X
Lysozyme C	X		X	X	X	X	X	X
Lysozyme G	X		X		X	X	X	X
Cathelicidin *	X		X		X	X	X	X
**Protein (sub)families with a possible secondary function as AMP**								
A1P/Antitrypsin	X	X		X	X	X	X	
Adrenomedulin	X		X		X	X	X	X
Apelin						X		
Beta-amyloid	X	X	X		X	X	X	X
Amylin/Calcitonin					X	X		X
Chemokine	X	X			X	X	X	X
Chrombacin/Secretogranin	X		X	X	X	X	X	X
Cofilin	X	X	X	X	X	X	X	X
DBI (Diazepam Binding Inhibitor)	X		X	X	X	X	X	X
Enolase	X		X	X	X	X	X	X
GADPH (Glyceraldehyde phosphate dehydrogenase)	X	X	X	X	X	X	X	X
Granulin	X		X	X	X	X	X	X
Histone H2A—Acipensin 6 region	X	X	X	X	X	X	X	X
Histone H2A—Hipposin region	X	X	X	X	X	X	X	X
Histone H2B	X		X		X	X	X	X
Histone H3	X	X	X		X	X	X	X
IBP (Insulin-like growth factor binding protein)	X	X	X	X	X	X	X	X
Neuropeptide W	X				X	X		
Neuropeptide YY	X	X	X		X	X		X
PACAP (Pituitary adenylate cyclase-activating polypeptide)	X		X		X	X	X	X
POMC (Pro-opiomelanocortin)	X					X		X
Proenkephalin	X		X		X	X		
Thymosin	X					X		X
Vasostatin	X	X	X		X	X	X	X
Cystatin/Kininogen *	X		X		X	X	X	X

^G+T^ species with genomic and transcriptomic data. ^T^ species with transcriptomic data only. * BLAST matches against UniProt records.

**Table 2 toxins-16-00150-t002:** Results of directional selection analyses using the FADE method. AA (target amino acid). Site (alignment position directionally evolving). Bias (substitution bias). Bayes factor (above 100 is considered significant). Results are only shown for those candidate AMPs in which signatures of directional selection were detected.

AA	Site	Bias	Bayes Factor	AA Site Composition	Inferred Substitutions History
**Lysozyme C** (clade 1)					
F	54	25.09	122.56	F8, Y15	Y->F(4)
H	107	34.01	293.40	A1, D13, E1, H3, N4, S1	D->A(1)E(1)H(2), N->D(1)H(1)S(1)
**Lysozyme C** (clade 2)					
S	10	34.13	234.10	S15	-
**Cathelicidin**					
H	754	21.06	447.56	H28, I1, L12, V1	H->L(1), L->I(1)V(1)
I	360	36.83	124.09	I12, R35	R->I(2)
**Histone H2A—Hipposin region**					
D	41	41.49	165.23	D3, E43	E->D(2)
**Histone H2B**					
G	39	29.83	100.12	A23, G7, S1	A->G(3)S(1)
**PACAP**					
N	28	36.74	150.08	N6, T5	T->N(2)

## Data Availability

This study did not generate any new sequence data but used data already available from previous studies (See Section 5.1). Amino acid sequences of identified candidate antimicrobial peptides (AMPs) are available as Appendix A.

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
