# Peer review of "Diversity and Molecular Evolution of Antimicrobial Peptides in Caecilian Amphibians"

_toxins, 2024, doi:10.3390/toxins16030150_

Round 1
Reviewer 1 Report
Comments and Suggestions for Authors
Nowadays, in-silico methods are the third pillar of scientific research. In the study under consideration, the authors utilize digital tools to identify antimicrobial peptides in caecilian amphibians. The methods are well justified, and the results are thoroughly analyzed and commented on. Such findings enrich our knowledge of antimicrobial peptides and are valuable starting points for studies aiming at AMP-based antibiotic development.
Minor comments:
1. The introduction would be even more complete if the authors added a short text regarding the mechanisms of action of the antimicrobial peptides, as the authors say that their results would be helpful in elucidating AMP models of action. Furthermore, AMPs can also be classified according to their mechanism of action.
2. Table 4 extends over several pages and is difficult to read.
3. The font size of the labels "a)" "b)" "c)" in Figures 9, 10 and 11 are disproportionate compared to the rest of the text in the figures and the text of the article as a whole.
Reviewer 2 Report
Comments and Suggestions for Authors
The manuscript is focused on the search for AMP candidates derived from caecilians, a type of amphibians that has been hardly studied before in this context. The authors take advantage of genomics and transcriptomics available data to find candidate sequences for AMPs, a clever approach and very useful as a tool to face the problem of the rising antibiotic resistances. The work is very well designed and the results are clearly structured and presented, which allows the reader to follow the text fluently. The interest and novelty of this work is out of any doubt, given the fact that caecilians are very poorly studied group of animals, from the AMP point of view.
The approach takes advantages of recently developed tools, such as Alpha Fold, and it is an excellent start point for the design of in vitro screening of the candidates in future studies.
I don't find any major considerations on the paper contents, project methodology, results presentation, and hardly any ortographic issue (except for two closing parenthesis missing in lines 42 and 95).
Reviewer 3 Report
Comments and Suggestions for Authors
In the current article, the authors present an interesting study on antimicrobial peptides from caecilian amphibians, employing blast analysis of the amphibians' genome with known AMP servers. The objective of the research is fascinating. Unfortunately, the paper is disorganized and presents a significant challenge in terms of understanding. The article appears lengthy, lacking a cohesive flow and resulting in confusion for the reader. Unfortunately, I think substantial revisions are required before contemplating any additional revisions. I am providing just a few examples to illustrate.
The division of one result into two figures (Table 1 and Figure 4) could be streamlined into a single figure, as could Table 2 and Figure 2.
Figure 3, in my view, maybe adequately represented through text alone, reducing complexity.
Table 4, representing predicted activity, may benefit from reconsideration due to its perceived lack of real-world significance.
I also seek clarification on the relevance of the extensive structural comparison in sections 2.5 and Figures 9-11 to the present study.
In summary, a more concise version of the paper, achieved by judicious removal of unnecessary sections, would enhance its overall quality and readiness for publication.
Reviewer 4 Report
Comments and Suggestions for Authors
Authors send their manuscript entitled ‘Diversity and molecular evolution of antimicrobial peptides in 2 caecilian amphibians’ for revision to TOXINS journal.
Authors in their work described antimicrobial peptides in side of investigate the diversity and molecular evolution of the AMPs present in the genomes and transcriptomes of eight species of caecilians.
Manuscript needs revision and patr that should be improved are listed below:
1. Page 4 line 124 why Authors wrote ‘(see Supplementary Table S4 therein), which’ that Table 4 not as Table 1, so it is mentioned as first in manuscript text, moreover in Supplementary file there is no Table 4S.???? there are only Tables S1-S3. Please check and add respective data.
2. In Table 1 on page 4 there are 7 AMP families and 10 Protein/peptide families with a possible secondary function as AMP, so there is wrong number ‘13’ in sentence on page 7 line 168. also page 19 line 402. Please check and correct.
3. Page 6 Figure 2 why only 3 species were placed in this Figure , but Authors ealier in mentioned 8 Caecilians species, it is why these 3 have species with genomic and transcriptomic data???
4. page 8 Figure 4 there are only 3 species not 8 as ealier in Fig 1. it is why these 3 have species with genomic and transcriptomic data???
Unfortunately I was unable to open and see files attache as Supplementary files: file ‘Sequences_of_candidate_AMPs’ they were saved as .FASTA files
Reviewer 5 Report
Comments and Suggestions for Authors
The paper entitled « Diversity and molecular evolution of antimicrobial peptides in caecilian amphibians » (toxins 2807178) reports the identification of potential antimicrobial peptide (AMP) sequences from available genomic and/or transcriptomic datasets of eight caecilian species. The 331 candidate sequences were compared to those of two anuran species (X. laevis and B. bombina) and to that of the lungfish P. amectens. In silico analysis suggests that up to 162 of these AMPs could potentially develop antimicrobial activity.
In the absence of experimental data, we do not learn much new thanks to this study, but this work has the merit of highlighting potential AMPs in amphibian species that have been little studied. The paper is well written, and the analysis appears to have been performed correctly. I do not have any particular objections to the publication of this article in Toxins.
I would just like to suggest that the authors standardise the presentation of Tables 1 and S1, perhaps by using colours, to make it easier for the reader to find their way around. About this Table1, I do not understand why I. kotahoensis does not express G-type lysozyme and defensin, although Table S1 shows the contrary ? Similarly, Table 1 shows that T. natans expresses PACAP, although Table S1 shows the contrary ? This should be clarified.
I also would like to suggest to the authors to remove Fig. 3. In fact, the case illustrated by the schematic representation in « c », is not demonstrated by sequence alignement, and the caption of the figure invites the reader to refer to the text ! So, the text sends the reader to the figure, and the figure sends the reader back to the text !
Table 4 is difficult to use to understand what characterises AMPs in different species. It would be helpful to the reader if Table 4 was presented on a single page, if it’s possible to do so.
Line 250, it is confusing to refer to Table 3 here, since this Table concerns only cofilin and lysozyme, and not CGRP and IAPP. This confusion appears also in the Method section Line 726.
Line 302 « we analysed the 179 peptides… » line 305 « Of the 180 peptides… »
Reviewer 6 Report
Comments and Suggestions for Authors
The authors analyzed the genome of eight species of caecilian amphibians, characterized their molecular evolution and the in silico antimicrobial potential of the detected antimicrobial peptides (AMPs).
Of the detected AMPs, the authors analyze their phylogenetic sequences and real-world applications against some of the most common microorganisms on a case-by-case basis.
The study is very detailed, materials and methods are adequately illustrated, and the pathophysiological hypothesis of the work is interesting. The discussion is equally insightful and it is consistent with the results of the study. Conclusions are detailed.
English is well written and the text flows fluently.
The pictures and tables are clear in content and form.
All in all, the work is meticulous, precise and appropriately supported in its exposition.
Round 2
Reviewer 3 Report
Comments and Suggestions for Authors
The authors have satisfactorily addressed all of my concerns, and I have no remaining issues. I extend my thanks and congratulations to the authors.